# An Evaluation of the Impact of Illegal Dumping of Solid Waste on Public Health in Nigeria: A Case Study of Lagos State

**DOI:** 10.3390/ijerph20227069

**Published:** 2023-11-16

**Authors:** Eloho Beatrice Ichipi, Mpinane Flory Senekane

**Affiliations:** Department of Environmental Health, Faculty of Health Sciences, University of Johannesburg, Johannesburg 2028, South Africa; ichipibeloho@gmail.com

**Keywords:** environmental degradation, illegal dumping, public health, solid waste, solid waste management, waste generation

## Abstract

The illegal dumping of solid waste is a prevalent issue in Nigeria, affecting all states. Lagos State, in particular, faces waste disposal problems that stem from urbanization, negative public perception towards waste management, insufficient waste disposal education, poor waste disposal practices, and the disposal of waste at illegal and unauthorized sites. This situation is further exacerbated by inadequate municipal solid waste (MSW) collection rates, making it challenging to combat illegal dumping effectively. To align with Sustainable Development Goal (SDG) 11 of Sustainable Cities, which seeks to minimize negative environmental effects associated with managing MSW by 2030, this study aims to assess the environmental and health impact of illegal dumping of solid waste in Lagos State. The objectives of this study include assessing people’s attitudes towards illegal dumping, evaluating the extent of environmental degradation resulting from illegal solid waste dumping in Lagos, and assessing the health risks associated with exposure to illegally dumped solid waste in Lagos State, as well as determining if illegal dumping contributes to the diseases prevalent in Lagos State. This study will use a quantitative approach to collect data from study participants on demographics, educational background, waste management, and environmental and health issues using computer-assisted self-administered questionnaires (Google Forms). This study will also use observations and photographs of illegally dumped waste in communities and major illegal dumpsites in the study area to provide valuable information to complement the questionnaire responses. By combining both approaches, the study will be able to capture both numerical data and people’s behaviors and attitudes associated with illegal dumping. This study will use a mixed sampling method consisting of cluster sampling and convenience sampling, focusing on communities impacted by illegal dumping in Lagos State. The sample size for this study will be 100. The questionnaire for this study will be administered via a Google Forms link that will be shared through various online platforms, enabling participants to access and complete the questionnaire from any location with internet access. To ensure representative participation, as part of the informed consent form on the initial page of the online questionnaire, a screening question that requires participants to indicate if they live or work in the study areas or have been exposed to the impact of illegal dumping will be included. Only those who meet these criteria will be able to proceed with the study. The primary data obtained will be carefully analyzed using the Statistical Package for the Social Sciences (SPSS) version 28. The study’s results will highlight the importance of the linkage between illegal dumping, environmental degradation, and diseases prevalent in Lagos State, which could inform policymakers and relevant departments in developing effective strategies to improve public health.

## 1. Introduction

This study protocol serves as a blueprint for a research endeavor that outlines the systematic step by step approach and methodology for conducting a comprehensive study on the environmental and health impact of illegal dumping of solid waste in Lagos State, Nigeria. In developing the research methodology for this study, well-established and proven research methods described and tested in published academic studies were employed. Specifically, this study drew from the work of [1,2] who employed similar methods in their studies.

Over the past decades, environmental concerns have increasingly become a global issue, with the adverse effects of poor waste management practices being a critical challenge for society worldwide [3,4]. Improper waste disposal, including illegal dumping, has resulted in environmental degradation, posing a significant threat to the health and well-being of individuals and communities [5]. Despite global efforts to properly manage solid waste, illegal dumping remains a significant contributor to environmental degradation, particularly in developing countries, with African nations bearing the highest burden [6,7].

In developing countries, including Nigeria, the challenges of managing solid waste have been compounded by rapid population growth, increased urbanization, poverty, and inadequate government support [8]. There exists a real-time correlation between urbanization, population growth, industrialization, and waste generation [9]. Lagos State, as one of the fastest-growing urban centers in Nigeria, faces a significant challenge in effectively managing its solid waste, with illegal dumping becoming a widespread practice. This practice can lead to severe environmental and health consequences, including water, soil, and air contamination, and the spread of diseases [6,10]. As a result, illegal dumping has been characterized as a “wicked act” due to its complexities, including intractable, open-ended problems and rights-based and legal issues [10]. Illegal dumping, a significant contributor to poor waste management, leads to many types of environmental pollution in communities, costing cities worldwide millions yearly for cleanup [11]. The continuous increase in urban populations is a precursor to an increase in waste generation rates and, consequently, indiscriminate solid waste disposal, contributing to environmental challenges such as water, soil, and air pollution; blocked water drains, resulting in flooding and water stagnation in drainage systems; and waste items that favor water-borne diseases such as cholera and vector-borne diseases such as malaria and dengue [7].

Therefore, effective waste management should prioritize public health and environmental protection, aligning with the integrated sustainable waste management (ISWM) model. This model suggests that waste must be managed in a way that safeguards public health and the environment, aligning with the Sustainable Development Goals (SDGs), particularly SDG 11 (Sustainable Cities and Communities), emphasizing sustainable waste management and the mitigation of negative environmental impacts [12].

### 1.1. Illegal Dumping in Developing Countries (Africa)

Waste generation in African cities is among the lowest in the world, but the demand for waste services is not met due to the unavailability of reliable, geographically comprehensive data and information [6]. As a result, it is very challenging to plan, evaluate, and monitor local, national, and regional waste management systems, which leads to improper management of solid waste and the multiple facets of waste management challenges faced by all African countries [13]. According to the UNEP, available records show that 13% of MSW generated in sub-Saharan Africa is made of plastic, and 57% is organic waste, with a collection rate of 55%, the majority of which is currently dumped illegally, undermining Africa’s efforts to meet the Sustainable Development Goals (SDGs).

### 1.2. The Challenges Faced in Dealing with Illegal Dumping in Nigerian Cities

In most Nigerian cities, solid waste management poses a significant environmental concern [14]. Urbanization-related rapid population growth in Abuja metropolitan areas, the rapid increase in waste generation rates that resulted from this, the high cost of waste disposal, the exhaustion of landfill space, and the difficulty of obtaining new disposal sites that results in open dumping are the issues that have made it difficult for the state and local environmental protection authorities to effectively and efficiently manage the growing volumes of waste generated by the city [15]. According to Kadafa [15], improperly disposed solid waste has become an environmental and health hazard in areas like the suburbs of Abuja, within which open illegal dumps are a common sight, with waste management authorities blaming it on the attitude and educational level of the populace; the populace, on the other hand, attributes it to the authorities’ infrequent collection, with some areas not receiving waste management services. The Abuja Environmental Protection Board’s (AEPB) characterization of municipal solid waste shows that organic waste (50%), paper (25%), and plastics, including water sachets (18%), account for most of the solid waste generated in Abuja, posing environmental and public health risks if discarded illegally [15]. A study conducted in Kubwa-Abuja to investigate the hazardous influence of leachate on the environment as a result of poor waste disposal indicated that diarrhea, dysentery, and malaria were the most common ailments in the population [16].

Due to the city’s growing population, as well as its industrial and economic development, Akure’s solid waste generation has risen steadily over time, from an estimated 60,000 metric tonnes annually in 1996 to 75,000 metric tonnes in 2006 [17]. As a result, numerous studies have attempted to assess the potential effects. The study by Ojo (2020), carried out in Akure, revealed a 62.5% collection rate of municipal waste by municipal authorities while the remaining 37.5% was subjected to illegal dumping and open burning, with the highest proportion of the solid waste being organic (27%) and plastics (18%). Illegal dumping and open burning would present environmental and public health risks to the city due to the well-known environmental and health issues with organic and plastic waste. Groundwater contamination, unpleasant odors, uncontrolled dumping, and risk to public health are the main issues associated with indiscriminate waste disposal. According to a study of the public health implications of solid waste management in Akure, measles, diarrhea, malaria, and typhoid fever where the most common illnesses linked to poor environmental sanitation [17].

Lagos, one of Africa’s megacities and Nigeria’s economic and financial center, is increasingly undergoing urbanization and industrialization, placing a burden on its finances and infrastructures, resulting in inadequate service delivery in sectors such as environmental health and waste management [18]. The state is thus faced with the challenge of the successful management of solid waste generated by millions of residents by ensuring a safe environment through the collection, transportation, disposal, and resource recovery of municipal solid waste [19]. Collection of waste from inner cities and informal settlements is a major challenge; the inefficiency frequently encountered in the waste collection can be attributed to factors such as bad roads, the city’s traffic congestion, and the nature of vehicles used for collection [20]. Due to these inadequacies, a vast amount of waste is uncollected as well as dumped in unauthorized places, resulting in an estimate of up to 80,000 tonnes of waste generated in a period of six months dumped in 100 illegal dumpsites in Lagos [21]. A study conducted on two communities where major landfills are in Lagos revealed constant bouts of malaria as the major ailment suffered by residents, as well as skin irritation and chest-related problems [22].

### 1.3. Problem Statement

Illegal dumping is a significant environmental degradation issue that poses health risks to humans, animals, and the environment. Despite efforts by countries to manage solid waste properly, illegal dumping still accounts for a major portion of all waste generated globally, with the burden disproportionately affecting African countries. In Nigeria, illegal dumping presents a significant health risk to both the environment and human health [15,23,24], with uncollected waste being a common sight on the streets of Lagos and other parts of the country [25]. Poor waste disposal education, inadequate waste management practices, and the prevalence of illegal dumping sites are some of the most pressing issues facing waste management authorities in Nigeria. Another contributing factor to this problem is the negative perception of waste management among the public [26] and the inability of waste management authorities to collect waste timeously. This research initiative uses Lagos State as a case study to assess the impact of illegal dumping of solid waste on public health, with a particular focus on its implications for environmental degradation and health risks.

### 1.4. Research Question

How much of an impact does illegal dumping of solid waste have on the environment and human health in Lagos State?

### 1.5. Aims and Objectives

The aim of this study is to assess the environmental and health impact of illegal dumping of solid waste in Lagos State.

The objectives are stated below.

To establish the attitude of people living in Lagos State towards illegal dumping.To assess the extent of environmental degradation resulting from illegal solid waste dumping in Lagos State.To evaluate the health risks associated with exposure to illegally dumped solid waste in Lagos State.To evaluate if illegal dumping contributes to the diseases prevalent in Lagos State.

### 1.6. The Gap in Research

There has not been research using quantitative data to measure the impact of illegal dumping on environmental and human health risks and their impacts on the most prevalent diseases in Lagos State.

## 2. Methods and Materials

### 2.1. Experimental Design

This research will utilize a cross-sectional quantitative study design to gather data. A mixed sampling method, consisting of cluster sampling and convenience sampling, will be used to collect the data required for this study.

### 2.2. Materials and Equipment

Survey Questionnaire 

The survey questionnaire is presented in Table A1 of the Appendix A.

The primary data collection tool for this study is a web-based survey questionnaire created using Google Forms.The questionnaire includes closed-ended and open-ended questions covering demographic information, educational backgrounds, waste management practices, environmental and health perceptions, and experiences related to illegal dumping.

Data Analysis Software

Statistical Package for the Social Sciences (SPSS) version 28 will be used for data cleaning, analysis, and visualization.

### 2.3. Research Setting

The study setting for this research will be online amongst residents living in communities that are highly impacted by illegal dumping in Lagos State. Figure 1 on page 5 shows the map of Lagos State.

### 2.4. Detailed Procedure

#### 2.4.1. Study Population

The study population will consist of individuals who have been directly or indirectly affected by illegal dumping, as well as those who have witnessed or experienced adverse effects on their health or the environment. Participants aged 18 years and older of all sexes and socio-economic backgrounds will be included. To ensure that the sample population is representative of the areas with the most significant issues related to illegal dumping, the study will be conducted in five selected Local Government Areas (LGAs) in Lagos State, identified by the 2020 Lagos State Household Survey Report by the Lagos Bureau of Statistics (LBS) to be heavily impacted by illegal dumping [27].

#### 2.4.2. Sample Size Calculation

Using Epi Info 7.2, the sample size was computed using the following parameters: expected frequency of 48%, as determined in a study by Ike et al. (2018); 90% confidence level (CL); and 8% confidence limit (margin of error). Given the high population of people aged 18 years and older in Lagos and across all LGAs and the study design that involves the use of the internet, where anyone can enter the study at any time, the population size is set to be infinite, while the cluster size is set to five, given that the research participants will be recruited from five selected LGAs. Therefore, it was estimated that a sample size of 110 would be required.

#### 2.4.3. Sampling Strategy

This study will use a mixed sampling approach consisting of cluster sampling and convenience sampling. The five Local Government Areas (LGAs) with the highest impact of illegal dumping will be selected using secondary data obtained from a 2020 Lagos State household survey report by the Lagos Bureau of Statistics (LBS) [27]. These LGAs include Alimosho, Ikeja, Ojo, Ikorodu, and Lekki, all situated in the Lagos East and Lagos West districts. In the second stage, participants who reside or work in the affected communities will be recruited through social media platforms using convenience sampling. To ensure representative participation, as part of the informed consent form on the initial page of the online questionnaire, a screening question that requires participants to indicate if they live or work in the study areas or have been exposed to the impact of illegal dumping will be included. Only those who meet these criteria will be able to proceed with the study. In the questionnaire, participants will be asked to provide information about their residential and work areas in questions 5 and 6, respectively. In question 7, participants will be asked to indicate which of these areas is specifically impacted by illegal dumping. This information will allow for the classification of study participants according to each of the Local Government Areas (LGAs). By collecting data on participants’ residential and work areas, as well as their identification of impacted areas, it will be possible to extrapolate information and classify the study participants based on their association with the selected LGAs.

#### 2.4.4. Inclusion and Exclusion Criteria

Inclusion criteria:Participants living in Lagos State.Participants must be 18 years and older.Participants are willing to participate in the study and can give consent.

Exclusion criteria:Participants living outside Lagos State.

#### 2.4.5. Data Collection

The quantitative data for this study will be collected using a survey questionnaire by means of Google Forms (Appendix A). A crucial feature of Google Forms is the ease with which vast amounts of information are captured from many people through a simple web form that immediately saves the data to Google Sheets [28]. The web-based, self-administered questionnaire (Google Forms) will be written in English and used to collect primary data from study participants on demographics, educational backgrounds, waste management, and environmental and health issues.

#### 2.4.6. Pilot Study

Prior to the start of the main data collection, a pilot study with ten participants will be conducted to allow the researchers to identify and address any issues with the questions. The participants will be administered the same questions as the primary study participants.

#### 2.4.7. Data Analysis

To ensure accurate information, entered data will be cleaned by eliminating missing values, checking for duplicates, and finding any other anomalies in the dataset. The data collected for this study will be analyzed using the Statistical Package for the Social Sciences (SPSS) version 28 tool. The descriptive analysis will be used to summarize the data by calculating the measures of central tendency (mean, median, and mode) and measures of dispersion (range, standard deviation) to describe the distribution of the data. This analysis will help to describe the characteristics of the study population, the prevalence of waste disposal practices, and the extent of the impact of illegal dumping of solid waste on public health in Lagos State. Inferential analysis will be used to test the hypotheses generated from the research questions and establish relationships between variables. The findings of the analysis will be visualized using tables and graphs.

#### 2.4.8. Data Management Plan

The datasets obtained from this study will be transformed into digital formats and stored securely in the UJ Data Repository (DR) for a duration of five years, to ensure their preservation and safeguarding for future reference or utilization. Should there be a requirement to utilize these data for academic or research purposes in the future, they will be subjected to further ethical assessment and approval. After the completion of the fifth year, all copies will be permanently erased. 

#### 2.4.9. Validity and Reliability or Trustworthiness

In this study, several measures will be taken to ensure the validity of the findings. Firstly, the questionnaire will be pre-tested among a sample of the target population to ensure it is clear and easy to understand through the pilot study. Secondly, cluster and convenience sampling will be used to ensure that participants are selected from areas that are most impacted by illegal dumping. Finally, data analysis will be conducted using appropriate statistical methods to ensure that the findings are accurate and reliable. To ensure the reliability of the study, the questionnaires administered to primary study participants will first be administered to pilot study participants. Any issues identified during the pilot testing will be addressed to ensure the questionnaires are reliable.

## 3. Ethical Considerations

### 3.1. Informed Consent

The process of obtaining informed consent for this research will differ from the traditional approach of conducting a face-to-face interview. This is because the research will be conducted using an online web questionnaire, and therefore, the informed consent will also be obtained online. To ensure that the principle of informed consent is upheld, the information letter and consent form will be integrated into the initial pages of the web questionnaire. Participants in the study will be required to confirm their understanding of the information provided and indicate whether they consent to taking part in the study. They will be given the option to agree or decline to participate by checking a box. If they choose to participate, their response will be recorded as informed consent. If they decline to participate, they will be unable to proceed with the study. Participants will also be informed that they may withdraw their consent prior to submission of data but beyond this point, withdrawal of consent will not be possible.

### 3.2. Privacy and Confidentiality

To ensure participant confidentiality and promote free expression, the questionnaire will be self-administered and completed anonymously. Therefore, the questionnaire will not request identifying information such as names, addresses, or contact details.

### 3.3. Risks and Benefits

No study participant will experience any kind of risk or harm because of taking part in this research. Equal opportunity will be given to everyone to participate in the study. The questionnaire will take roughly 20 min to complete.

### 3.4. Gatekeeper Permissions

Approval will be obtained from the Departmental Research Committee, Higher Degree Committee, and Ethics Research Committee in the Faculty of Health Sciences, University of Johannesburg, who will give their approval before this study can commence, as well as approval from the Lagos Waste Management Authority (LAWMA).

Timeline (see Table 1).

## 4. Expected Results

Results obtained will be presented using narrative reports, tables, and graphs. Below are templates of tables (Table 2, Table 3, Table 4, Table 5, Table 6, Table 7 and Table 8) that demonstrate how the study results will be represented.

Environmental Degradation. See Table 4, Table 5 and Table 6.

**Table 4 ijerph-20-07069-t004:** Types of Waste Materials Commonly Found in Illegal Dumping Sites.

Types of Waste Materials	Frequency (N)	Percentage (%)
Plastic waste		
Glass waste		
Metal waste		
Electronic waste/e-waste		
Other		

**Table 5 ijerph-20-07069-t005:** Environmental Effects of Illegal Dumping.

Environmental Effects	Frequency (N)	Percentage (%)
Pollution of surface water		
Contamination of underground water		
Land and soil pollution		
Air pollution		
Loss of urban beauty		
Flooding		
Other		

**Table 6 ijerph-20-07069-t006:** Reasons for Illegal Dumping.

Reasons for Illegal Dumping	Frequency (N)	Percentage (%)
Lack of adequate waste disposal facilities		
Poor waste disposal habits among the general population		
Lack of awareness about the negative impact of illegal dumping		
Lack of enforcement of waste disposal regulations by the government		
High cost of using authorized waste disposal facilities		
Lack of education on proper waste disposal practices		
Cultural or social norms that prioritize immediate convenience over long-term environmental impact		
Other		

Health Risks. See Table 7 and Table 8.

**Table 7 ijerph-20-07069-t007:** Primary Health Risks Associated with Exposure to Illegal Dumping.

Health Risks/Problems	Frequency (N)	Percentage (%)
Vector-borne diseases (malaria, dengue, Lassa fever, etc.)		
Skin problems (infection and irritations)		
Gastrointestinal illnesses (cholera, diarrhea, food poisoning)		
Respiratory tract infection (flu, sinus, bronchitis, cough, pneumonia, asthma)		
High blood pressure (hypertension) and heart problems		
Hazardous chemical exposure (exposure to electronic, chemical, and medical waste)		
Other		

**Table 8 ijerph-20-07069-t008:** Diseases Associated with Illegal Dumping.

Diseases Associated with Illegal Dumping	Frequency (N)	Percentage (%)
Malaria		
Skin problems		
Cholera/Diarrhea		
Respiratory tract infection (e.g., flu, sinus, bronchitis, cough, pneumonia, asthma)		
High blood pressure (hypertension) and heart problems		
Typhoid fever		
Other		

## 5. Conclusions

The outcome of this research is expected to contribute valuable insights to the body of knowledge on illegal dumping by providing valuable insights into people’s attitudes toward illegal dumping in Lagos State, the levels of environmental degradation caused by illegal solid waste dumping in Lagos State, the significant health risks associated with exposure to illegally dumped solid waste in Lagos State, and the contribution of illegal dumping to the prevalence of certain diseases in Lagos State. Overall, these findings may result in recommendations to help mitigate the negative effects of illegal dumping and to influence policy and practice in waste management, environmental protection, and public health.

## Figures and Tables

**Figure 1 ijerph-20-07069-f001:**
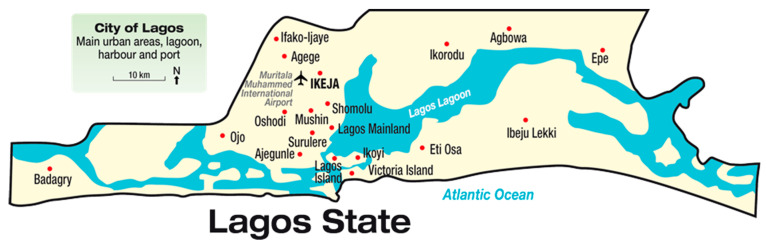
The map of Lagos State.

**Table 1 ijerph-20-07069-t001:** Timeline for the research study.

Activities 2022–2023	Nov.	Dec.	Jan.	Feb.	Mar.	Apr.	May.	June	July	Aug.	Sept.	Oct.	Nov.
Submission for reviews and approvals													
Data collection													
Data analysis													
Write-up													
Examination													

The colours serve as timelines in which the study will be conducted.

**Table 2 ijerph-20-07069-t002:** Demographic variables of participants.

Variable	Frequency (N)	Percentage (%)
Gender - Female - Male Age of participants - 18–30 - 31–40 - 41–50 - 51–60 - 60 and above Education of participants - Tertiary education - Secondary education - Primary education - Never went to school - Others Marital status of participants - Single - Married - Divorced/Separated - Widowed Residential Area - Alimosho - Ikeja - Ojo - Ikorodu - Lekki Work Area - Alimosho - Ikeja - Ojo - Ikorodu - Lekki Areas impacted by illegal dumping - Residential area - Work area		

**Table 3 ijerph-20-07069-t003:** Attitude Towards Illegal Dumping.

Question	Frequency (N)	Response
Do you believe that illegal dumping of solid waste is a problem in Lagos State?		
How often would you say illegal dumping of solid waste occurs in your residential or work area?		
How often is your community cleaned, e.g., street swept and illegally dumped waste collected?		
How likely are you to file a complaint against someone who disposes of their waste in an unauthorized place?		
Have you personally engaged in illegal dumping of solid waste?		
Are there signs or media adverts about the dangers of illegal and indiscriminate dumping in your community?		
Do you know there are risks/problems associated with illegal dumping?		
If you answered yes to the question above, what are the associated risks/problems?		
Do you think more needs to be done to inform the public about the risks associated with illegally dumping solid waste?		

## Data Availability

No new data were created or analyzed in this study. Data sharing is not applicable to this article.

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
