# Peer review of "An Evaluation of the Impact of Illegal Dumping of Solid Waste on Public Health in Nigeria: A Case Study of Lagos State"

_ijerph, 2023, doi:10.3390/ijerph20227069_

Round 1
Reviewer 1 Report (Previous Reviewer 2)
Comments and Suggestions for Authors
The introduction has improved it, as well as the references. The methodology section was updated and it was improved, especially a step-by-step approach of how the study will be carried out. It was important to be included the expected result section, as well as templates of tables. Considering all this improvement, in my opinion, now warrants publication in IJERPH as Study Protocol.
Author Response
Reviewer 1 has warranted the manuscript to be published in its current state. There are no comments. We just corrected the American/ United Kingdom spelling to be consistent in the document. This is highlighted in yellow in the document.
Reviewer 2 Report (New Reviewer)
Comments and Suggestions for Authors
The paper presents a study structure to evaluate of the impact of illegal dumping of solid waste on public health in Lagos. It seems that a detailed questionnaire will be conducted and some valuable information will be obtained on all the aspects of illegal dumping. Some minor editing for the language would be helpful. It could be published after that.
Comments on the Quality of English LanguageEnglish is acceptable, moderate revision would be helpful.
Author Response
Good day
Reviewer 2 did not specify areas where language editing is required, however, we went through the whole document to check for any errors and made correction.
Regards
Dr. Senekane
This manuscript is a resubmission of an earlier submission. The following is a list of the peer review reports and author responses from that submission.
Round 1
Reviewer 1 Report
Comments and Suggestions for Authors
- Please see attachment for details

Minor editing of English language required. References should be provided using MDPI format.
Author Response
attachment for details

Reviewer 2 Report
Comments and Suggestions for Authors
it is a well-designed work, with good foundations and good methodological construction. However, it is incomplete because it does not apply the methodology.
However, Considering that article as Study Protocol I suggest approved it. It is suggested to resubmit the work after having the results, because the publication potential will be high, depending on the quality of the analysis and conclusions.
Author Response
attachment for details.
